# Enhancing Genetic Gains in Grain Yield and Efficiency of Testing Sites of Early-Maturing Maize Hybrids under Contrasting Environments

**DOI:** 10.3390/genes14101900

**Published:** 2023-09-30

**Authors:** Baffour Badu-Apraku, Adamu Masari Abubakar, Gloria Boakyewaa Adu, Abdoul-Madjidou Yacoubou, Samuel Adewale, Idris Ishola Adejumobi

**Affiliations:** 1International Institute of Tropical Agriculture, PMB 5320, Oyo Road, Ibadan 200285, Nigeria; a.abubakar@cgiar.org (A.M.A.); s.adewale@cgiar.org (S.A.); i.adejumobi@cgiar.org (I.I.A.); 2Council for Scientific and Industrial Research—Savanna Agricultural Research Institute (CSIR-SARI), Tamale 00233, Ghana; gloriaboakyewaa@yahoo.com; 3Crop Breeding Department, National Institute of Agricultural Research of Benin/CRA, Cotonou 01BP884, Benin; abdoulmadjidou.yacoubou@yahoo.com

**Keywords:** *Striga*-infested, drought, low N, stability, repeatability, maize

## Abstract

The major challenges of maize production and productivity in Sub-Saharan Africa (SSA) include *Striga hermonthica* infestation, recurrent drought, and low soil nitrogen (low N). This study assessed the following: (i) accelerated genetic advancements in grain yield and other measured traits of early-maturing maize hybrids, (ii) ideal test environments for selecting early-maturing multiple-stress tolerant hybrids, and (iii) high-yielding and stable hybrids across multiple-stress and non-stress environments. Fifty-four hybrids developed during three periods of genetic enhancement (2008–2010, 2011–2013, and 2014–2016) were evaluated in Nigeria, The Republic of Benin, and Ghana under multiple stressors (*Striga* infestation, managed drought, and Low N) and non-stress environments from 2017 to 2019. Under multiple-stress and non-stress environments, annual genetic gains from selection in grain yield of 84.72 kg ha^−1^ (4.05%) and 61 kg ha^−1^ (1.56%), respectively, were recorded. Three mega-environments were identified across 14 stress environments. Abuja was identified as an ideal test environment for selecting superior hybrids. The hybrid TZdEI 352 × TZEI 355 developed during period 3 was the most outstanding under multiple-stress and non-stress environments. On-farm testing of this hybrid is required to verify its superior performance for commercialization in SSA. Considerable progress has been made in the genetic improvement of early-maturing maize hybrids for tolerance of multiple stressors and high yield. The identified core testing sites of this study could be used to enhance the testing and selection of promising hybrids.

## 1. Introduction

Maize is a major staple food crop ranked first among cereals in terms of grain production and second in economic value after rice (*Oryza sativa* L.) [1,2]. In Sub-Saharan Africa (SSA), maize is essential for food security and provides almost half of the dietary calories and protein intake for about 50% of the population. By the mid-century, maize demand in SSA is predicted to increase threefold because of population growth and dietary changes [2,3,4,5,6,7]. Depite the importance of maize in SSA, the yield potential is rarely maximized [8,9] because of recurring stressors, particularly drought [10], low soil fertility [11,12], and *Striga hermonthica* parasitism (Giant Witchweed) [9], which increasingly limit maize yield. Consequently, yield is low in SSA, averaging a little over 2.0 Mg ha^−1^ on farmers’ fields [1,13].

An estimated area of about 40% under maize production in SSA is affected by occasional drought while 25% is prone to recurrent drought [10,12,14], with yield losses varying between 17–60% [15]. However, if moisture stress occurs a few days before and during anthesis, yield losses could be as high as 90%, [16]. Besides recurrent drought, a major challenge for small-scale farmers in SSA is the declining soil fertility characterized by poor nitrogen (N) content [17]. Fertilizer use in SSA ranges between 5 and 10 kg ha^−1^ which is far lower than the 100 and 96 kg ha^−1^ in Asia and Latin America, respectively [18]. Consequently, yield losses of between 10–50% have been reported in SSA [19]. Another major challenge of maize production is the damage caused by the parasitic plant, *S. hermonthica*, which threatens food security and puts at risk the profit of smallholder farmers [20]. Staple food losses due to *Striga* infestation are estimated at 4.1 million metric tons, amounting to $7 billion yearly [9,21]. Maize yield reduction resulting from *Striga* parasitism varies between 0 and 100% depending on the severity of infestation, crop stage, varietal type, weather conditions, and soil health and nutrient levels [22]. Complete yield losses have been reported under severe *Striga* infestation within SSA [20,23,24], forcing farmers to abandon their farms. To meet predicted global demand, improving maize to enhance tolerance/resistance to *Striga* and drought and for efficient nitrogen use is urgently required [25,26].

Maize improvement in West and Central Africa (WCA) is addressed by the International Institute of Tropical Agriculture (IITA) in collaboration with National Agricultural Research Partners (NARS). Through this and other strategic collaborative efforts, several high-yielding hybrid maize and open pollinated varieties have been developed and released, many of which are adapted to a broad range of agro-ecological zones within the sub-region. Generally, under non-stress conditions, grain yield is highly correlated with maturity. However, if stressors (e.g., drought or low-N) are encountered during the growing season, the positive relationship between yield and maturity is lost, and invariably, early maturing hybrids out-yield late maturing hybrids. Consequently, early and extra-early maize hybrids have been targeted largely to drought-prone environments in SSA which are consistently characterized by drought stress and/or short growing seasons [27]. Early and extra-early maize ensure early harvests which are used to fill the hunger gap in July/August when all food reserves are exhausted after the long dry period [28]. Therefore, these hybrids have rapidly become the primary components of production systems in the harsh drylands of WCA as they ensure higher productivity, shorter life cycle, and high response to applied nutrients, as well as improved acceptability to farmers [29].

Drought stress, *Striga* parasitism, and low soil N are the major constraints in increasing maize production and productivity in the savanna agro-ecological zones of WCA. As a result, maize varieties developed for the *Striga*-prone areas of WCA must possess both drought and low N tolerance. Therefore, the primary goal of the early-maturing maize improvement program for biotic and abiotic stress tolerance at IITA is to improve grain yield under conditions of low soil nitrogen, drought, and *S*. *hermonthica* infestation. Routine studies on gains from selection after long periods of genetic improvement from a breeding program offer great potential to enhance breeding strategies by assessing the efficiency of past improvement programs and suggesting future selection directions to facilitate further improvement. Such studies are routinely embarked on for most staple crops including maize [30,31,32]. In Southern and Eastern Africa (ESA), genetic yield gain estimates in maize hybrids have established that yields have improved by 109.4, 32.5, and 141.3 kg ha^−1^ annually from 2000–2010 under optimal, drought, and Low N conditions, respectively [31]. Similarly, in early-maturing Open Pollinated Varieties (OPVs), the authors reported yield gains of 109.9, 29.2, and 192.9 kg ha^−1^ yr^−1^ from 1999–2011, respectively under optimal, drought, and Low N environments. In WCA, Ref. [29] reported genetic enhancement in the maize grain yield of OPVs to values of 44 and 67 kg ha^−1^ yr^−1^ under multiple stressors and optimal environments, respectively, during the period from 1995 to 2012 and attributed this to the availability of extra-early hybrids with additional days to anthesis, enhanced resistance to stalk lodging, and improved husk cover. In general, the stacking of genes for increased plant and ear heights, plant and ear aspects, husk cover, and increased ears per plant has resulted in enormous increases in maize grain yields under non-stress conditions [33]. However, information is unavailable detailing the breeding progress of the early-maturing hybrids generated during the three breeding periods of genetic enhancement under multiple-stress environments. This makes it difficult to determine the genetic gains from selection for grain yield under multiple-stress and non-stress environments.

The IITA Maize Improvement Program (IITA-MIP) has, during the period 2008–2016, developed several early-maturing hybrids, specially targeted to the savanna agro-ecologies and the second growing season in the forest agro-ecological zones of WCA. The nine years have been divided into three breeding periods based on the specific strategies used for maize genetic enhancement: 2008–2010 (Period 1), 2011–2013 (Period 2), and 2014–2016 (Period 3). The breeding strategies adopted in developing these hybrids have been described in detail by [34,35]. Fifty-four hybrids (18 each in periods 1, 2, and 3) have been selected for their superior performance in regional trials for the different breeding periods. There is, therefore, a need to evaluate the hybrids in field trials to ascertain whether the present rates of improvement will satisfy future production requirements.

The presence of genotype × environment interactions (G × E) has been demonstrated in the multi-environment experiments (METs), [36,37,38]. The selection of superior hybrids is made more difficult by the presence of considerable G × E. This justifies the need for extended hybrid testing in the target region over years in multiple environments before registration and release [39,40,41]. However, because of resource constraints and the difficulty of conducting trials under stressful environments, maize research programs of WCA are required to conduct evaluations in a few selected environments, mostly under non-stress conditions [42], occasionally limiting the reliability of the results. As a result, it is crucial to advance our knowledge and continually evaluate the effectiveness and representativeness of the test environments. This should facilitate the effective deployment of maize hybrids with high yield potential that are adapted to the contrasting stress and non-stress environments for increased adoption by farmers. The objectives of this study were as follows: (i) to determine the rate of genetic gain in grain yield of early-maturing maize hybrids developed by IITA during the 2008 to 2016 period under non-stress and multiple-stress environments, (ii) to identify high-yielding and stable hybrids across multiple-stress and non-stress environments, and (iii) to identify the ideal test environments for selecting early multiple-stress tolerant hybrids.

## 2. Materials and Methods

### 2.1. Development of Multiple Stress-Tolerant Early-Maturing Hybrids for the Genetic Gain Study

The IITA early-maturing inbred lines development program was started in 1994. The program aimed at developing early-maturing open-pollinated varieties, inbred lines, and hybrids with moderate to high levels of tolerance to *Striga* from TZE-W Pop DT STR C0, TZE-Y Pop DT STR C0, TZE Comp 5-Y C6, and TZE-W Pop × 1368 STR. The details of the methodology utilized for the development of the S_6_ inbred lines and synthetic varieties from each population have been described by [30]. Briefly, selected S_1_ lines from the diverse germplasm sources were advanced to S_4_ stages of inbreeding. Following each cycle of inbreeding, the lines were evaluated under artificial *Striga* infestation and induced moisture stress. At the S_4_ stage, 250–300 selected lines were crossed to a broad-based tester for estimation of general combining ability in test crosses as proposed by [35]. Based on the performance of the test crosses, 90–100 S_4_ lines were advanced to the S_6_ stage of inbreeding employing the pedigree selection scheme under artificial *Striga*-infested and moisture-stress environments [21]. Through this program, numerous S_6_ inbreds and synthetic varieties were extracted from these populations. Each of these populations possessed enhanced *Striga*-resistance and drought-tolerance, making them important sources of *Striga*-resistant inbred lines and synthetic varieties. However, the levels of resistance to *Striga* and tolerance to drought of the early maturing maize populations were not as high as desired. In 2007, a program commenced aimed at increasing the frequency of favourable alleles for tolerance to drought in the early-maturing maize populations using the S_1_ family recurrent selection scheme. This led to new generations of outstanding, early-maturing multiple-stress-tolerant populations, combining improved levels of drought tolerance, resistance to *Striga*, and tolerance to low N [30].

A panel of multiple-stress-tolerant early yellow and white endosperm hybrids was assembled from the early hybrids developed for high resistance to *Striga,* tolerance to drought, and low N during 9 consecutive years from 2008 to 2016. In total, 54 hybrids developed during three breeding periods (2008–2010, 2011–2013, and 2014–2016) were used in this study. The hybrids were selected for their outstanding performance in regional variety trials in WCA, with many of them sharing the same female parents regardless of the year of origin. Each breeding period has 18 hybrids. Information on the hybrids used in this study is shown in Appendix A.

### 2.2. Management of Field Trials

The 54 maize hybrids were evaluated in Nigeria, The Republic of Benin, and Ghana across 14 stress environments (managed drought, *Striga*-infestation, and low soil N) and 21 non-stress environments (high N, *Striga*-free, and rain-fed) conditions between 2017 and 2019. The location and year combination was regarded as the environment. Descriptions of test environments are presented in Table 1.

Trials were evaluated using an α-lattice design (9 entries in 6 blocks) in three replicates. Each plot consisted of two rows, 4 m in length, spaced 0.75 m apart with within-row spacing of 0.40 m. Three seeds were sown per planting hole and thinned to two plants per stand, two weeks after planting, to attain a final population density of 66,666 plants per hectare (ha^−1^). Trials under induced drought stress were conducted at Ikenne, Nigeria in the dry seasons of 2017 to 2019. Ikenne is characterized by Eutric nitrisol [43,44]. For each year, managed drought trials were planted in mid-November so that flowering occurred in mid-January when the incidence of rainfall was insignificant. At the time of planting, NPK 15:15:15 fertilizer was applied at a rate of 60 kg N, 60 kg P, and 60 kg K ha^−1^. Three weeks later, an additional 60 kg N ha^−1^ was added. During the first 25 days after planting (DAP), water was applied weekly using a sprinkler irrigation system that supplied 17 mm. After that, irrigation water was stopped until the crop reached maturity, forcing the maize plants to rely on the water reserve in the soil for growth and development.

The low N trials were conducted at Ile-Ife and Mokwa in Nigeria throughout the growing seasons of 2017 to 2019. Nitrogen was depleted from the soil by regularly planting maize for many consecutive years and clearing the field of stover after harvest. The soil in Mokwa is a luxisol [23] with 0.27, 0.035, and 0.48% organic C, organic N, and organic P contents, while the soil at Ile-Ife is an alfisol [23] with 0.084% organic N. Prior to planting, soils were sampled annually and N concentration was determined at the IITA soil laboratory in Ibadan. The Technicon AAII Auto Analyzer, Kjeldahl digestion, and colorimetric determination were used to measure the total N in the soil. Additional fertilizer application was done at 2WAP to increase the total N in the soil to 30 kgha^−1^. Additionally, single superphosphate (P_2_O_5_) and muriate of potash (K_2_O) were applied at 60 kg ha^−1^.

The hybrids were also evaluated for yield potential under *Striga*-infested conditions during the growing seasons of 2017 to 2019 in Ghana, the Republic of Benin, and Nigeria. In Nigeria, the hybrids were evaluated at Mokwa and Abuja (*Striga*-prone locations) from June to October, each year. In the Republic of Benin and Ghana, the hybrids evaluation was done in the planting season of 2017 at Ina and Nyankpala, respectively. In each of these locations, fumigation of the fields was performed with ethylene gas 7 days before planting to promote the suicidal germination of *Striga* seeds. Field infestation with *Striga* was performed following the procedure described by [45]. Additionally, trials were conducted in 21 environments under non-stress growing conditions across the three countries during the 2017 to 2019 growing seasons. At planting, 60 kg ha^−1^ N, P, and K were applied. Top-dressing with N fertilizer was done at 4 WAP with Urea 46:0:0 at 60 kg N ha^−1^. Under *Striga*-free conditions, 30 kg ha^−1^ of N, P, and K were applied using compound fertilizer NPK 15–15–15 between 21 and 25 DAP. Weeds were controlled using herbicides and/or manually.

### 2.3. Traits Measured

During the trials evaluation for both stress and non-stress environments, data were collected as described in the Table 2.

### 2.4. Data Analyses

Firstly, the data were analysed for each environment for a broad-sense heritability (*H*) estimate of grain yield as follows:(1)H=σg2σg2+/+σe2/r
where $\sigma_{g}^{2}$ is the genetic variance, and $\sigma_{e}^{2}$ is the error variance; r is the replicates per environment.

Any trial with heritability estimates of less than 0.30 was eliminated from further analyses. As a result, 13 multiple-stress and 21 non-stress environments were subjected to analysis of variance (ANOVA) using PROC GLM in SAS 9.4 [46]. An ANOVA was performed for each stress environment, across stresses, and non-stress environments. In the ANOVA, all factors except genotypes were considered random effects. Means separation was achieved using the standard error. Variance components were estimated using the restriction maximum likelihood method in SAS MIXED [46].

Repeatability of the traits under multiple-stress and non-stress environments was computed using the following formula:(2)$$R=\sigma_{g}^{2}/\sigma_{g}^{2}+\frac{\sigma_{g\times e}^{2}}{e}+\frac{\sigma_{e}}{re}$$
where $\sigma_{g}^{2}$ = variance of genotype, $\sigma_{g\times e}^{2}$ = genotype × environment interaction and $\sigma_{e}^{2\;}\;$= residual variance; e = number of environments, and r = number of replicates.

The regression analysis was used to determine the relationship between measured traits of the maize hybrids and year of origin across stress and non-stress environments. The mean grain yield (dependent variable) was regressed on the year of origin (independent variables) to obtain regression coefficients (b values) across stress and non-stress environments. To estimate genetic gain per year, the b value was divided by the intercept and expressed as a percentage. Furthermore, the relationship between grain yield under multiple-stress and non-stress environments was visualized for each breeding period using clustered column-line in Excel software (v.2016).

A multiple trait base index (MI) comprising YLD, EPP, ASI, PASP, EASP, SGR, SD, and ESP under stress and grain yield under non-stress environments was used to select the best 15, middle 15, and worst 5 hybrids [33]. The mean values of the traits with significant effects from ANOVA were standardized. A positive MI value indicated tolerance/resistance and negative values indicated susceptibility. The equation below was used to compute the MI.
(3)$$MI=\left(2+{YLD}_{STR}\right)+{YLD}_{NSTR}+EPP-ASI-EASP-PASP-SGR-SD8-SD10-\left(0.5\times ESP8\right)-\left(0.5-ESP10\right)$$

The mean grain yield data of the top 15, middle 15, and worst 5 hybrids evaluated across the 13 stress and 21 non-stress environments were subjected to GGE biplot analysis to decompose the G × E interactions using the GGE biplot v. 4.0 [36,47,48] available at www.ggebiplot.com (accessed on 20 December 2022). The entry number of each of the environments is shown in Appendix A. The biplot was plotted using the first two principal components (PC1 and PC2). The data had the following properties (transformation = 0, standardization = 0, and centering = 2). The biplot was based on SVP 2, making it suitable for visualizing the relationships among environments. For relationships among hybrids, the biplot was based on SVP 1. This provided information on adaptability of the hybrids to different environments, the stability of the hybrids in the contrasting environments, and the identification of the mega-environments.

## 3. Results

### 3.1. Analysis of Variance across Stress and Non-Stress Environments

Results of the combined analysis of variance (ANOVA) for grain yield and other traits across the multiple stress environments showed highly significant (*p* < 0.001) mean squares for environments (E), periods, hybrids (period), hybrids (period) × E interactions, and E × period interactions for all measured traits, except for the period mean squares for ear rot, and E × period mean squares for root lodging and stay green characteristics (Table 3). Similarly, under non-stress environments, significant mean squares were observed for environments (E), periods, hybrids (period), hybrids (period) × E interactions, and period × E interactions for all traits except period mean squares for days to silk (Table 4). In the combined ANOVA, repeatability estimates of the traits varied from 0.50 for RL to 0.92 for DA under the stress environments, and from 0.51 for EPP to 0.96 for PHT under non-stress environments.

### 3.2. Genetic Enhancement in Grain Yield of the Hybrids in the Three Breeding Periods under Stress and Non-Stress Environments

The grain yield under multiple stressors was 2244 ± 357.1 kg ha^−1^ for hybrids developed from 2008 to 2010 and 2531 ± 425.7 kg ha^−1^ for hybrids developed from 2011 to 2013. Similarly, hybrids developed from 2014 to 2016 had a mean grain yield of 2796 ± 256.5 kg ha^−1^. Generally, mean grain yield was high for hybrids developed during period 3 with a genetic gain of 4.05% yr^−1^. Across non-stress environments, mean grain yield was 4345 ± 427.6 kg ha^−1^ for hybrids bred during periods 1, 4779 ± 383.0 kg ha^−1^, and 4876 ± 406.1 kg ha^−1^ for hybrids developed during periods 2 and 3, respectively, with an annual genetic gain of 1.56% yr^−1^ (Table 5 and Table 6). The average rate of yield increase measured was 84.7 kg ha^−1^ yr^−1^ under stress and 65.0 kg ha^−1^ yr^−1^ under non-stress environments (Table 6). Generally, under stress environments, a highly significant (*p* < 0.001) increase in grain yield was observed for the period 3 hybrids compared to those developed during periods 1 and 2. Similarly, across non-stress environments (*p* < 0.05) significant gains in grain yield was observed for hybrids developed in period 3 compared to those of periods 1 and 2 (Table 5). The increases in grain yield under contrasting stressors were accompanied by considerable increases in EPP, PHT, and EHT. Additionally, significant increases in grain yield in stress environments accompanied reduced ASI, SD8 and SD10, and improved PASP and EASP. Furthermore, no significant increases or decreases for DA and DS were observed in the present study in both stress and non-stress environments.

The individual grain yield performance of the 54 early maize hybrids (18 from each period) in both stressful and non-stressful environments were assessed and compared (Figure 1). The hybrid performance under stressful environments is represented by the horizontal lines, while their performance in non-stressful environments is represented by the vertical lines. The results show a clear distinction between hybrids of the three breeding periods. Fifty-four hybrids, 18 from each period, competed for a total of 18 points under stress and non-stress environments, i.e., 3 hybrids, 1 from each period, competed for a point. Under stress environments (horizontal points), period 3 hybrids scored a total of 9 points (50%), and period 2 hybrids scored 8 points (44%), while hybrids from period 1 had 1 point (6%). Under non-stress environments, hybrids from period 3 earned a total of 10 points (56%), those from period 2 had a total of 7 points (39%), and hybrids from period 1 had 1 point (5%).

### 3.3. Performance and Stability of Early-Maturing Maize Hybrids of Three Breeding Periods across Environments

Using the MI under multiple-stress conditions, the best 15, middle 15, and worst 5 hybrids were selected based on the means of grain yield and other agronomic traits. The base index values ranged from −19.7 for period 1 (TZEI 63 × TZEI 87) × (TZEI 59 × TZEI 108) to 17.3 for period 3 (TZdEI 352 × TZEI 355). Under multiple stress environments, grain yield varied from 1683 kg ha^−1^ for (TZEI 31 × TZEI 63) to 3808 kg ha^−1^ for TZdEI 352 × TZEI 355 and 3473 kg ha^−1^ for TZEI 31 × TZEI 18 to 5628 kg ha^−1^ for TZdEI 352 × TZEI 355 under non-stress environments. Similarly, mean grain yield for the stress conditions ranged from 551 to 2886 kg ha^−1^ under induced drought stress, 2311 to 4248 kg ha^−1^ under low N, and from 1352 to 4252 kg ha^−1^ under *Striga* infestation. Accordingly, the GGE biplot analysis revealed that hybrid 25 (TZdEI 352 × TZEI 355), developed during period 3, has the highest yield and it is the most stable across stress and non-stress environments (Figure 2). Hybrid TZdEI 352 × TZEI 355 had superior performance compared to the other hybrids. In contrast, the lowest-yielding hybrid, Hybrid 6 (TZEI 31 × TZEI 18), was from Period 1.

### 3.4. Assessing the Core Testing Sites for Selecting Early Multiple-Stress Tolerant Hybrids

In this study, the location-by-year combination was treated as an environment. Therefore, three mega-environments were identified. The first mega-environment consisted of E1 (Ile-Ife low N, 2017), E6 (Ikenne drought, 2017), E7 (Ikenne drought, 2018), E8 (Ile-Ife low N, 2018), E11 (Mokwa low N, 2018) and E14 (Ikenne drought, 2019). The second mega-environment comprised E3 (Abuja *Striga*-infested, 2017), E4 (Ina *Striga*-infested, 2017), E9 (Abuja *Striga*-infested, 2018), E10 (Mokwa *Striga*-infested, 2018), and E1 (Ile-Ife low N, 2019) while E2 (Mokwa *Striga*-infested, 2017) and E5 (Nyankpala *Striga*-infested, 2017) constituted the third mega-environment (Figure 3). The discriminating power and representativeness view of the GGE biplot of the target environments is presented in Figure 4. Environments with small angles with AEA are more representative of the mega-environment than those that have large angles with it. In the present study, environments E11, E6, E14, and E8 had short vectors. In contrast, environments E9 (Abuja *Striga*-infested, 2017) and E3 (Abuja *Striga*-infested, 2018), with long vectors and small angles with AEA, were identified as ideal test environments for hybrid discrimination and allowed selection of superior genotypes in contrasting environments. Environments E9 and E3 were highly discriminating and representative test environments. In addition, a high correlation existed between these environments.

The significant mean squares for grain yield and other studied traits observed for test environments, periods, and hybrids across the contrasting environments implied that the test environments were unique and that significant differences existed among the hybrids of the different periods. These results agree with those of [24,36]. The presence of significant hybrids (period) × E and Period × E mean squares for grain yield and other measured traits under the contrasting environments signified the existence of differential responses in hybrids. This necessitated the identification of high-yielding and stable hybrids across the contrasting environments. These results are consistent with the report of [49,50]. This emphasized the need for testing hybrids in several environments across years before recommendations for the commercialization of the hybrids are made. Additionally, the non-significant period mean squares observed for ear rot and E × period mean squares for root lodging across stress environments and period mean squares for days to silking across non-stress environments demonstrated consistency in the trait expression of the hybrids of the three different breeding periods. Most measured traits had high repeatability (i.e., ≥0.60) under the different stress and non-stress conditions, indicating that each of the test environments had a significant influence on the measured traits.

## 4. Discussion

In a breeding program, it is important to determine the level of progress made throughout a specific period of genetic improvement. It is striking that, under the contrasting environments, significant improvements were achieved in grain yield and other measured traits of early-maturing maize hybrids developed across the three breeding periods. The genetic gain and average rate of increase in grain yield obtained in this study was higher compared to the 1.33% yr^−1^ reported by [33] across *Striga* infestation, induced drought stress, and low soil nitrogen. Similarly, the results of the present study revealed higher genetic gains from selection compared to the gains in grain yield of 1.93% yr^−1^ reported by [34]. This is not surprising because hybrids respond more favorably to selection compared to open-pollinated varieties [51]. It is, therefore, of particular interest that IITA-MIP has, during the last two decades, focused more on the development and commercialization of hybrids compared to open-pollinated varieties. Increased ears per plant, plant, and ear heights were accompanied by considerable increases in grain yield under contrasting stressors. Similarly, reduced anthesis-silking intervals, *Striga* damage ratings at 8 and 10 WAP, and improved plant and ear aspects accompanied by significant increases in grain yield under stress environments, were observed. Additionally, no significant increases or decreases for days to anthesis and days to silking were observed in the present study under both stress and non-stress environments. These are very interesting results because, during the development of the hybrids used for this study, IITA-MIP’s MI that involved EPP, ASI, EASP, PASP, SD8, SD10, ESP8, ESP10, and SGR was used for the selection of the early-maturing stress tolerant/resistant hybrids. These results agree with the findings of [49]. Furthermore, the non-significant increases or decreases in days to anthesis and silking observed in this study under stress and non-stress environments indicated that there were no differences in maturity among the hybrids developed during the three breeding periods.

The comparative performance of grain yield of the early maize hybrids, in both stress and non-stress environments, showed a clear distinction between the hybrids of the three breeding periods. The early hybrids from the third period scored exceptionally well in both stress (56%) and non-stress (50%) environments. These results confirmed that the hybrids from period 3 outperformed those from periods 1 and 2 in both contrasting environments. This suggested that, during the three breeding periods, significant advancement had been achieved in developing outstanding hybrids with improved levels of resistance to stressful environments.

Under multiple-stress conditions, the best 15, middle 15, and worst 5 hybrids were selected based on the means of grain yield and other agronomic traits using MI ranging from −19.7 for period 1 (TZEI 63 × TZEI 87) × (TZEI 59 × TZEI 108) to 17.3 for period 3 (TZdEI 352 × TZEI 355). Mean grain yield for the stress conditions ranged from 551 to 2886 kg ha^−1^ under managed drought, 2311 to 4248 kg ha^−1^ under low N, and from 1352 to 4252 kg ha^−1^ under *Striga* infestation. This demonstrated that under *Striga*-infested environments, the performance and responses of the hybrids to selection were greater. This may be because IITA-MIP’s selection for maize inbred lines with enhanced tolerance to *Striga*-prone environments has been the main emphasis for almost two decades in the early and extra-early maize breeding program. Out of the 15 hybrids with the best MI, 9 (60%) were bred in period 3, while 3 (20%) each were bred in periods 2 and 3. The hybrids with positive MI yielded above mean performance of 2500 and 3000 kg ha^−1^ in stress and non-stress environments, respectively. Additionally, the hybrids with positive MI possessed higher grain yield, delayed flowering date and senescence, improved plant and ear aspects, higher plant height, increased ears per plant, decreased *Striga* damage syndrome ratings, and reduced emerged *Striga* plants compared to those with negative MI. Compared to their susceptible counterparts, both tolerant and resistant hybrids had less yield reduction. The average grain yield in multiply stressful conditions was 39% lower than the average grain yield of non-stress conditions.

The GGE biplot is an invaluable tool for identification of the best genotypes across multiple test environments. In the biplot display, the first two PCs explained a comparatively higher percentage of the total variation (44%). This study revealed that the GGE biplot was effective in identifying superior candidates under contrasting environments by dissecting the overall variation among the hybrids. The double-arrowed (blue) line in the GGE biplot separated hybrids that yielded above average and those that yielded below average. As a result, the yield decreased as the hybrid moved further to the left of the double-arrowed line, while it increased as the hybrid yield moved further to the right of the double-arrowed line. Regarding the stability of the hybrids, the longer the projection of a hybrid onto the single-arrowed line, the lower the stability of the hybrid and vice versa [37]. Therefore, hybrid 25 (TZdEI 352 × TZEI 355), developed during period 3, was the most stable and highest yielding across stress and non-stress environments. This indicated that hybrid TZdEI 352 × TZEI 355 possessed favourable alleles that contributed to the observed superior performance compared to the other hybrids. This hybrid could be recommended for further testing in on-farm trials for consistency in performance and commercialization in SSA. Contrarily, the lowest-yielding hybrid, Hybrid 6 (TZEI 31 × TZEI 18), was from Period 1.

Representativeness of the environment refers to the capacity of a test location inside a mega-environment to accurately represent the mega-environment, while the discriminating power of an environment relates to the ability of environments to measure and identify an ideal test environment. The main objective of the mega-environment analysis is to understand the pattern of interaction between genotypes by environment (GE) within a target region and select test environments that effectively identify superior genotypes for a mega-environment. This assessment is required for the investigation of the possibility of dividing the target region into mega-environments, which would allow GE to be used to leverage the genetic basis of genotype adaptation to a particular environment. Consequently, it is possible to reduce the selection response within a mega-environment and improve total yield within a target environment [52,53,54]. According to [48], test environments should be divided into three types. The first group is environments with low genotype discrimination and should not be chosen for testing genotypes. The second group is environments with a high potential for genotype discrimination and representative of the mega-environments that are close to the ideal and should be selected for superior genotype selection. The third group involves environments with high genotype discriminating ability but do not represent the mega-environment, which could be utilized for unstable genotype evaluation. In this study, the classification of Ikenne and Ile-Ife as the first mega-environment for two consecutive years under low N and drought stress was not surprising because the two environments belonged to the same agroecological zone as described previously (Table 1). This confirmed that these locations provided similar information about the genotypes. These results suggested that prospective early-maturing hybrids chosen in one of these locations in a particular stress environment would also be suitable for production in other locations in various stress environments. Similarly, Mokwa *Striga*-infested, and Nyankpala *Striga*-infested environments constituted the third mega-environment, indicating that these locations provided similar information about the hybrids. Additionally, Mokwa could also be considered as an independent research environment and could be classified as a special environment because it is part of mega-environments II, III, and I. As a result, Mokwa might not be considered while selecting test environments or for choosing superior hybrids because of its situation in Nigeria’s agroecological zone between the woodland savanna and the Guinea savanna.

In assessing the discriminating power and representativeness view of the environments in the biplot, the cycle is the AEA whose direction is shown by the red arrow [48]. The environments that possess small angles with the AEA are the most representative of the mega-environment than those with large angles. The implication is that the cosine of the angle between an environment vector and the AEA are used to estimate the correlation coefficient between the hybrid values in that environment and the hybrid means across the environments [48]. Furthermore, a test environment marker that was close to the biplot origin, or one that had a short vector, indicated that genotypes in that environment performed similarly and, as a result, that environment offered little to no information regarding genotype differences. If the biplot does not adequately explain the majority of the GGE of the data, a short vector could also indicate that PC1 and PC2 did not adequately describe the environment. These environments could be excluded when choosing test environments. In the present study, environments E11, E6, E14, and E8 had short vectors. These environments were considered independent research environments, treated as unique, and could not be used as test environments. In contrast, environments E9 (Abuja *Striga*-infested, 2017) and E3 (Abuja *Striga*-infested, 2018), with long vectors and small angles with AEA, were identified as ideal test environments for hybrid discrimination, allowing superior genotype selection in a variety of environments. Environments E9 and E3 are the most discriminating and representative test environments and were highly correlated in their ranking of the hybrids. This implied that these environments provided similar information about the hybrids.

## 5. Conclusions

Substantial progress has been achieved in the genetic enhancement of early-maturing hybrids for multiple-stress tolerance/resistance and grain yield improvement. Under the contrasting environments used in this study, annual genetic gains from selection in grain yield of 84.72 kg ha^−1^ (4.05%) and 61 kg ha^−1^ (1.56%), respectively, were obtained. The hybrid TZdEI 352 × TZEI 355 developed during the third period was the most stable and highest yielding under the contrasting environments. This hybrid should be further tested in on-farm trials for commercialization in SSA to improve food security. Three mega-environments were identified across the 14 stress environments. The environments E9 (Abuja Striga-infested, 2017) and E3 (Abuja Striga-infested, 2018) were identified as the ideal test environments for hybrid discrimination and could be used to facilitate the testing and identification of promising hybrids.

## Figures and Tables

**Figure 1 genes-14-01900-f001:**
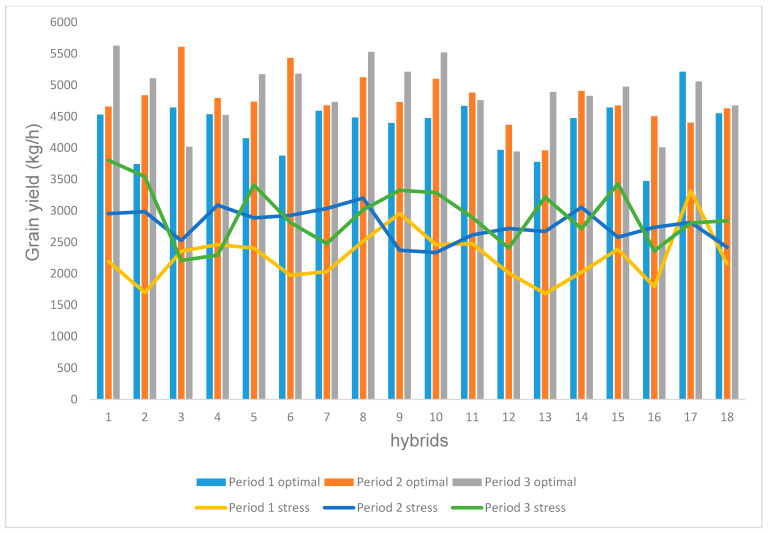
Comparative performance of the 54 early-maturing maize hybrids of the three breeding periods under multiple-stress (horizontal) and non-stress (vertical) environments.

**Figure 2 genes-14-01900-f002:**
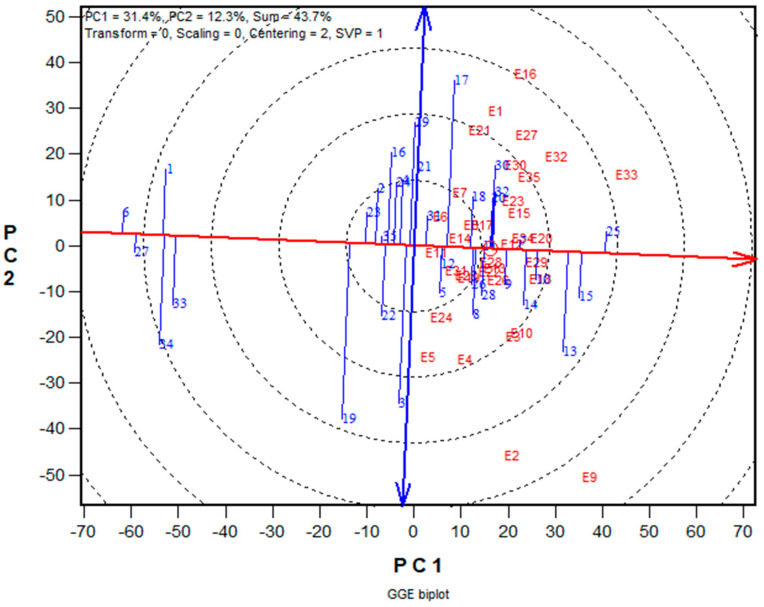
A “mean vs. stability” view of the genotype main effect plus genotype × environment interaction (GGE) biplot based on yield data of 35 early-maturing maize hybrids evaluated in 14 stress and 21 non-stress environments from 2017 to 2019 in West Africa. The red circle is the average environment abscissa (AEA), the red arrow is the direction of the AEA used to measure the stability of the hybrids and the double edge blue arrow is used to differentiate hybrids with higher yields from those with lower yields.

**Figure 3 genes-14-01900-f003:**
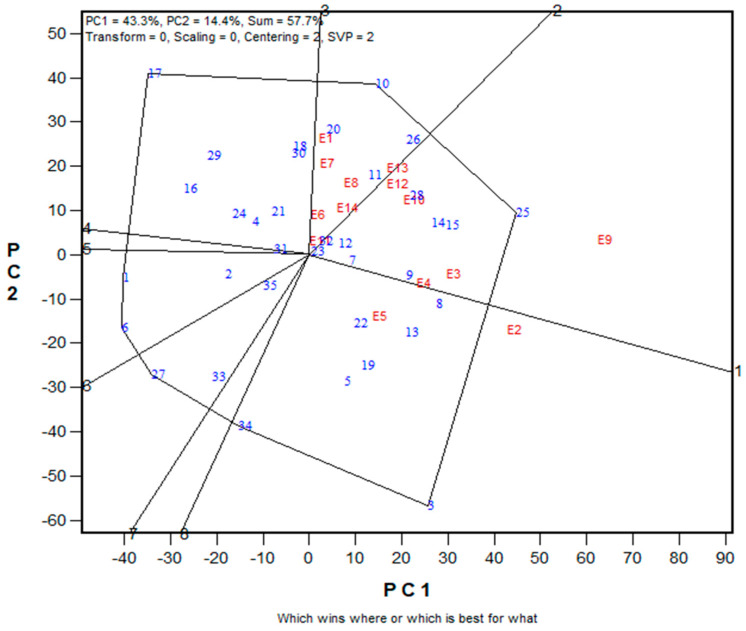
Polygon view of the genotype main effect and genotype by environment interaction (GGE) biplot of the 35 early-maturing maize hybrids evaluated in 14 stress environments in WA between 2016 to 2019.

**Figure 4 genes-14-01900-f004:**
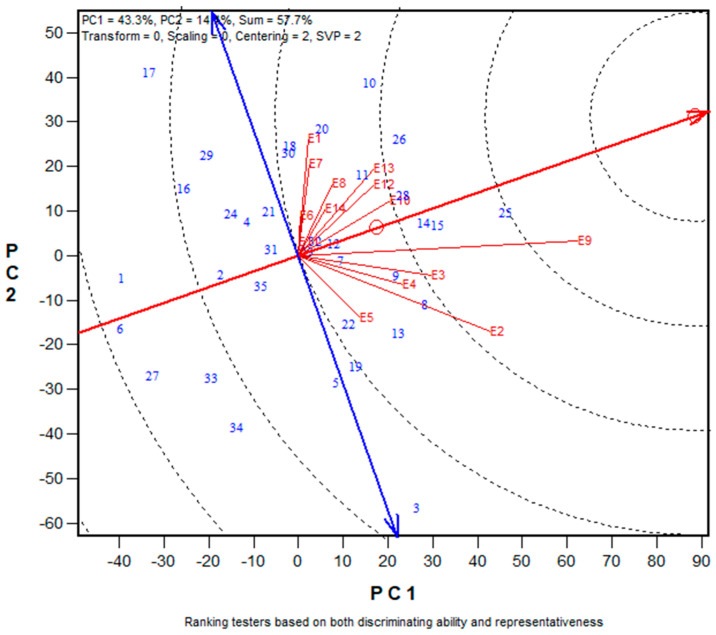
The ‘discriminating ability and representativeness’ view of the GGE biplot based on genotype × environment yield data of 35 early maize hybrids evaluated in 14 stress environments in WA between 2016 to 2019. The red circle is the average environment abscissa (AEA), the red arrow is the direction of the AEA used to measure the stability of the hybrids and the double edge blue arrow is used to differentiate hybrids with higher yields from those with lower yields.

**Table 1 genes-14-01900-t001:** Description of test locations used for evaluation of early maize hybrids in three breeding periods under multiple-stress and non-stress conditions in West Africa, 2017 to 2019.

Country	Location	Agro-Ecology	Latitude	Longitude	Altitude (m asl)	Soil Type	Average Annual Rainfall (mm)	Temperature (°C)
Nigeria	Abuja	* SGS	9°40′ N	7°29′ E	360	Lixisol	1389	25.7
Nigeria	Bagauda	SS	12°01′ N	8°19′ W	520	Arenosol	840	26.0
Nigeria	Ile-Ife	FT	7°28′ N	4°30′ E	244	Alfisol	1250	25.3
Nigeria	Ikenne	FT	6°35′ N	3°42′ W	60	Nitrisol	1264	27.0
Nigeria	Mokwa	SGS	9°18′ N	5°4′ E	457	Lixisol	1100	27.5
Nigeria	Zaria	NGS	12°00′ N	8°22′ E	640	Lixisol	1120	26.5
Ghana	Manga	SS	11°01′ N	0°16′ W	270	Lixisol	718	27.8
Ghana	Ejura	FT	7°38′ N	1°37′ E	90	Lixisol	1460	26.4
Benin	Angaradebou	SS	11°32′ N	3°05′ W	297	Lixisol	1000	28.0
Benin	Ina	NGS	9°58′ N	2°44′ W	297	Lixisol	1125	27.1

* SGS, southern Guinea savanna; SS, Sudan savanna; FT, forest-savanna transition zone; NGS, northern Guinea savanna.

**Table 2 genes-14-01900-t002:** Description of the measured traits of early maize hybrids of three breeding periods evaluated under stress and non-stress environments in WCA from 2017 to 2019.

Trait	Stage	Unit	Description
Days to 50% anthesis (DA)	Flowering	Days	Day count to 50% pollen shed from planting
Days to 50% silking (DS)	Flowering	Days	Day count to 50% silk emergence from planting
Anthesis-silking interval (ASI)	Flowering	Days	The time interval between 50% anthesis and silking
Plant height (PHT)	Post-flowering	Centimetre	The distance from the base of the plant to the first tassel branch for 10 representative plants per plot
Ear height (EHT)	Post-flowering	Centimetre	The distance from the base of the plant to the node carrying its upper ear for 10 representative plants per plot
Root lodging (RL)	Post-flowering	Percentage	Number of plants that were more than 30 degrees from the vertical expressed in percentage
Stalk lodging (SL)	Post-flowering	Percentage	Number of plants leaning more than 30 degrees from vertical or broken below the uppermost ear node expressed in percentage
Ears per plant (EPP)	Post-flowering	Numeric	Calculated by dividing the number of harvested ears by the number of plants harvested per plot
Ear aspect (EASP)	Harvest	Scale	Scored on a scale of 1 to 9, with 1 denoting ears that are tidy, uniform, big, and full, and 9 denoting ears with negative characteristics
Plant Aspect (PASP)	Post-flowering	Scale	Scored on a scale of 1–9 based on general phenotypic appearance of the plants in a plot, where 1 = excellent and 9 = very poor
Husk cover (HC)	Post-flowering	Scale	Scored on a scale of 1 to 9, with 1 denoting firmly packed husks that extended beyond the ear tip and 9 denoting exposed ear tips
Stay-green characteristic in drought and low-N fields (SGR)	Post-flowering	Scale	Scored on a scale of 1 to 9, where 1 represented nearly all green leaves and 9 represented nearly all dead leaves
Emerged *Striga* plant at 8 and 10 WAP in *Striga*-infested fields (ESP8 and ESP10)	Post-flowering	Count	The numbers of *Striga* plants that were counted at 8 and 10 WAP in the *Striga*-infested plots
*Striga* damage syndrome (SD8 and SD10)	Post-flowering	Scale	Scored on per plot basis on a scale of 1 to 9 where 1 = no damage, indicating normal plant growth and high resistance, and 9 = complete collapse or death of the maize plant
Grain yield (YLD)	Harvest	Kg/ha	Computed from the weight of the shelled grain adjusted to 80% shelling percentage and corrected for 15% moisture content [45]

**Table 3 genes-14-01900-t003:** Mean squares for grain yield and other measured traits of early maize hybrids of three breeding periods evaluated under Stress conditions in 13 environments in WCA from 2017 to 2019.

Source of Variation	df	Grain Yield	Days to Anthesis	Days to Silk	Anthesis-Silking Interval	Plant Height	Ear Height	Root Lodging	Stalk Lodging	Husk Cover	Ear Rot
Environment, E	12	149,343,973 **	948.67 **	1062.07 **	117.20 **	87,085.79 **	31,609.75 **	3786.66 **	15,261.20 **	168.34 **	7231.11 **
Block (E × Replicate)	195	2,333,911 **	5.63 **	9.71 **	2.01 **	550.31 **	261.01 **	64.55 **	71.92 **	0.75 **	36.22 **
Replicate (E)	26	4,332,085 **	11.91 **	19.83 **	3.28 **	1180.61 **	478.81 **	200.86 **	109.49 **	1.09 **	114.85 **
Period, P	2	49,981,471 **	18.89 **	104.79 **	31.02 **	16,823.09 **	2446.83 **	13.00 **	388.17 **	2.34 **	3.98
Hybrid, G	51	4,848,427 **	50.21 **	56.47 **	3.12 **	3052.92 **	907.66 **	76.51 **	209.36 **	4.85 **	84.27 **
E × G (P)	612	1,293,515 **	3.91 **	5.97 **	1.55 **	379.88 **	153.98 **	40.06 **	85.30 **	1.07 **	29.10 **
E × P	24	2,302,771 **	12.54 **	20.59 **	4.16 **	630.18 **	348.58 **	34.18	219.25 **	4.49 **	76.07 **
Error	1183	505,645	1.76	2.96	1.08	194.77	99.97	22.40	55.07	0.35	9.96
Repeatability		0.80	0.92	0.90	0.65	0.90	0.84	0.50	0.58	0.75	0.63
**Source of Variation**	**Ears/Plant**	**Ear Aspect**	**df**	**Plant Aspect**	**df**	**Stay Green Characteristic**	**Df**	***Striga* Damage (8 WAP)**	**Striga Damage (10 WAP)**	**Emerged *Striga* Plants (8 WAP)**	**Emerged** ***Striga* Plants** **(10 WAP)**
Environment, E	1.49 **	73.22 **	7	95.16 **	6	81.65 **	6	148.82 **	113.93 **	55,858.52 **	59,034.27 **
Block (E × Replicate)	0.04 **	2.35 **	120	0.96 **	105	1.22 **	105	1.61 **	1.66 **	468.98 **	696.12 **
Replicate (E)	0.09 **	3.42 **	16	2.10 **	14	3.06 **	14	4.71 **	6.80 **	2395.68 **	2999.63 **
Period, P	0.83 **	42.82 **	2	11.08 **	2	1.56 *	2	38.29 **	39.47 **	1955.59 **	2139.52 **
Hybrid, G	0.10 **	4.03 **	51	1.51 **	51	1.58 **	51	6.60 **	8.27 **	2033.06 **	1770.50 **
E × G (P)	0.04 **	1.37 **	255	0.75 **	306	0.78 **	306	1.50 **	1.55 **	601.41 **	714.33 **
E × P	0.08 **	3.74 **	10	1.42 **	12	0.63	12	4.89 **	5.74 **	1302.86 **	1627.54 **
Error	0.02	0.65	546	0.42	637	0.46	637	0.58	0.565373	299.07	426.30
Repeatability	0.71	0.74		0.58		0.57		0.79	0.82	0.70	0.60

*,** significant at 0.05 and 0.01 probability levels respectively.

**Table 4 genes-14-01900-t004:** Mean squares of grain yield and other measured traits of early-maturing maize hybrids in three breeding periods under non-stress conditions across 21 environments in WCA from 2017 to 2019.

Source of Variation	df	Grain Yield	Days to Anthesis	Days to Silk	Anthesis-Silking Interval	Plant Height	Ear Height	Root Lodging	Stalk Lodging	Husk Cover	Ear Rot	Ears/Plant	Ear Aspect	Plant Aspect
Environment, E	20	333,624,128 **	1323.95 **	1401.13	110.67 **	76,607.21 **	28,016.71 **	8094.50 **	509.87 **	143.53 **	5822.84 **	0.86 **	198.55 **	112.59 **
Block (E × Replicate)	315	1,271,631 **	3.48 **	4.20 **	0.59 *	366.80 **	188.17 **	59.18 **	13.11 **	0.56 **	30.43 **	0.02 **	0.74 **	0.72 **
Replicate (E)	42	3,873,073 **	8.89 **	10.12 **	0.97 **	688.04 **	519.08 **	230.44 **	15.15 **	1.32 **	199.97 **	0.02 **	2.63 **	1.18 **
Period, P	2	60,303,722 **	4.14	7.93 *	17.85 **	27,235.30 **	5519.18 **	480.23 **	80.51 **	5.86 **	124.33 **	0.22 **	31.49 **	30.17 **
Hybrid, G	51	11,497,047 **	68.69 **	79.19 **	2.62 **	7441.68 **	2028.35 **	261.37 **	46.53 **	4.17 **	86.92 **	0.03 **	4.68 **	7.06 **
E × G (P)	1020	1,544,388 **	3.33 **	3.92 **	0.73 **	307.92 **	164.31 **	69.99 **	15.21 **	0.81 **	22.78 **	0.02 **	1.02 **	0.70 **
E × P	40	2,284,330 **	4.91 **	5.31 **	0.92 **	449.77 **	253.33 **	133.40 **	19.47 **	1.37 **	33.17 **	0.03 **	2.59 **	1.44 **
Error	1911	512,583	1.84	2.17	0.50	173.88	99.70	36.44	10.28	0.37	8.61	0.01	0.41	0.39
Repeatability		0.88	0.95	0.95	0.78	0.96	0.93	0.74	0.68	0.82	0.75	0.61	0.81	0.91

*,** significant at 0.05 and 0.01 probability levels respectively.

**Table 5 genes-14-01900-t005:** Mean ± standard deviation of grain yield and other agronomic traits of early maize hybrids in three breeding periods were evaluated under 14 multiple-stress and 21 non-stress environments in WCA between 2017 to 2019.

Trait	Condition	Period		
		1	2	3
Grain yield, (kg/ha)	Stress	2244 ± 357.1	2531 ± 425.7	2796 ± 256.5
	Non-stress	4345 ± 427.6	4779 ± 383.0	4876 ± 406.1
Days to anthesis	Stress	54 ± 1.1	54 ± 1.0	54 ± 1.5
	Non-stress	54 ± 0.9	54 ± 1.3	54 ± 1.0
Days to silking	Stress	56 ± 1.1	56 ±1.0	55 ± 1.7
	Non-stress	55 ± 0.9	55 ± 1.5	55 ± 1.1
Plant height (cm)	Stress	139 ± 9.0	147 ± 10.2	149 ± 8.9
	Non-stress	156 ± 10.0	166 ± 9.9	163 ± 13.4
Ear height (cm)	Stress	65 ± 5.3	69 ± 5.4	68 ± 4.3
	Non-stress	73 ± 5.6	77 ± 4.6	78 ± 7.2
Plant aspect	Stress	5.3 ± 0.2	5.2 ± 0.3	4.9 ± 0.2
	Non-stress	4.5 ± 0.3	4.1 ± 0.4	4.3 ± 0.4
Ear aspect	Stress	4.9 ± 0.3	4.5 ± 0.4	4.5 ± 0.2
	Non-stress	3.8 ± 0.3	3.4 ± 0.3	3.6 ± 0.3
Ears/plant	Stress	0.7 ± 0.1	0.8 ± 0.0	0.8 ± 0.1
	Non-stress	0.9 ± 0.0	0.9 ± 0.0	0.9 ± 0.0
Anthesis silking interval	Stress	2.0 ± 0.4	1.9 ± 0.2	1.6 ± 0.3
	Non-stress	1.4 ± 0.2	1.1 ± 0.3	1.3 ± 0.1
Ear rot	Stress	5.8 ± 1.1	5.8 ± 1.8	5.7 ± 1.6
	Non-stress	5.1 ± 0.9	4.5 ± 1.2	4.5 ± 1.6
Husk cover	Stress	3.5 ± 0.2	3.3 ± 0.3	3.4 ± 0.5
	Non-stress	3.2 ± 0.2	3.0 ± 0.3	3.1 ± 0.3
Root lodging (%)	Stress	4.3 ± 1.6	4.1 ± 1.0	4.0 ± 1.6
	Non-stress	4.5 ± 2.6	4.9 ± 1.7	5.9 ± 2.2
Stalk lodging (%)	Stress	8.0 ± 1.6	7.5 ± 2.1	8.9 ± 2.8
	Non-stress	2.4 ± 0.9	2.2 ± 0.8	2.8 ± 1.0
Emerged Striga plants (8 WAP)	Stress	42 ± 10.2	39 ± 9.2	37 ± 11.9
	Non-stress	-	-	-
Emerged Striga plants (10 WAP)	Stress	50 ± 10.3	46 ± 8.3	46 ± 10.8
	Non-stress	-	-	-
Striga damage (8 WAP)	Stress	4.8 ± 0.6	4.1 ± 0.6	4.2 ± 0.6
	Non-stress	-	-	-
Striga damage (10 WAP)	Stress	5.3 ± 0.7	4.7 ± 0.6	4.8 ± 0.6
	Non-stress	-	-	-
Stay green characteristics	Stress	3.6 ± 0.4	3.4 ± 0.3	3.4 ± 0.3
	Non-stress	-	-	-

**Table 6 genes-14-01900-t006:** Relative genetic gain, coefficient of determination (R^2^), (a) slope, and (b) regression coefficients of grain yield and other agronomic traits of early maize hybrids in three breeding periods, evaluated under multiple-stress and non-stress environments in WCA between 2017 to 2019.

Trait	Relative Gain (% per Year)		R^2^		a		b	
	Stress	Non-Stress	Stress	Non-Stress	Stress	Non-Stress	Stress	Non-Stress
Grain yield, (kg/ha)	4.05	1.56	0.263	0.117	2093.7	4170.2	84.725 **	65.015 *
Days to anthesis	−0.03	0.1	0.002	0.014	53.81	53.53	−0.018 ns	0.052 ns
Days to silking	−0.17	0.02	0.033	0.001	55.99	54.98	−0.093 ns	0.013 ns
Anthesis silking interval	−3.35	−2.66	0.304	0.18	2.21	1.5	−0.074 **	−0.040 **
Plant height (cm)	1.27	1.18	0.183	0.146	136.26	152.75	1.731 **	1.796 **
Ear height (cm)	0.92	1.03	0.082	0.094	64.52	72.44	0.595 *	0.746 *
Root lodging (%)	−0.57	2.36	0.002	0.015	4.28	4.55	−0.025 ns	0.107 ns
Stalk lodging (%)	1.06	−0.71	0.008	0.003	7.73	2.55	0.082 ns	−0.018 ns
Husk cover	−0.26	−0.74	0.004	0.046	3.44	3.2	−0.009 ns	−0.024 ns
Plant aspect	−1	−1.29	0.241	0.155	5.41	4.63	−0.054 **	−0.060 **
Ear aspect	−1.61	−1.46	0.268	0.208	5.05	3.91	−0.081 **	−0.057 **
Ear rot	−0.14	−2.36	0	0.062	5.81	5.31	−0.008 ns	−0.125 ns
Stay green characteristics	−0.68	-	0.035	-	3.58	-	−0.024 ns	-
Striga damage (8 WAP)	−2.14	-	0.169	-	4.91	-	−0.105 **	-
Striga damage (10 WAP)	−1.95	-	0.143	-	5.48	-	−0.107 **	-
Emerged *Striga* plants (8 WAP)	−0.86	-	0.007	-	41.23	-	−0.355 ns	-
Emerged *Striga* plants (10 WAP)	−0.94	-	0.014	-	49.58	-	−0.466 ns	-
Ears/plant	1.49	0.46	0.224	0.16	0.72	0.89	0.011 **	0.004 **

*,** significant at 0.05 and 0.01 probability levels, respectively; ns, not significant.

## Data Availability

The datasets used in the present study are available at the IITA CKAN repository.

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
