# Peer review of "Enhancing Genetic Gains in Grain Yield and Efficiency of Testing Sites of Early-Maturing Maize Hybrids under Contrasting Environments"

_genes, 2023, doi:10.3390/genes14101900_

Round 1

Reviewer 1 Report

general comments

  1. What is the novelty of the research?
  2. References are not based on the MDPI structure.
  3. why do you use the self citation on your article? if you want get citation of your article, it is not good way!!!!! please remove or change all your self citation!!
  4. please separate result and discussion
  5. please follow the structure of the MDPI

Specific comments:

  1. Results are suitable for the article and the article's goal the article.
  2. The conclusion can be changed and focused on your novelty
  3. please add the version of the software you use it
  4. please check the df on table 2.
  5. quality of figures need to improve.

Author Response

RESPONSES TO REVIEWER’S COMMENTS

Reviewer 1

Comments and Suggestions for Authors

General comments

  1. What is the novelty of the research?

Response

The research is novel in the sense that it is the first report on genetic gains in early maize hybrids evaluated under multiple stress conditions. Prior to this report, earlier genetic gains studies had focused on the extraction of open-pollinated varieties. following repeated cycles of recurrent selection. Contrarily, in the present study hybrids developed from inbred lines have been used in estimating the genetic gains obtained after 9 years of hybrid development under multiple stress (Striga infestation, managed drought, and Low N) and non-stress environments.

   References are not based on the MDPI structure.

Response

The references have been modified to meet the MDPI structure as requested.

  1. why do you use the self citation on your article? if you want get citation of your article, it is not good way!!!!! please remove or change all your self citation!!

Response        

Please note that MDPI permits a maximum of 15% self-citation and the present manuscript has 12.96%. Nevertheless, we have further reduced self-citation to the barest minimum.

  1. please separate result and discussion

Response

The result and discussion have now been separated as suggested.

  1. please follow the structure of the MDPI

Response

The structure of the MDPI has now been followed as suggested

           Specific comments:

  1. Results are suitable for the article and the article's goal the article.

Response

Yes

  1. The conclusion can be changed and focused on your novelty

Response

The conclusion has been reviewed and modified to focus on the novelty of the research.

  1. please add the version of the software you use it

Response

The version of the software (GGE biplot v. 4.0) has been added as suggested.

  1. please check the df on table 2.

Response

The df on Table 2 has been checked thoroughly, The traits have not been consistently recorded in the same number of environments thus, explaining the variation in the df reported in the table. For the first part of the table comprising grain yield, days to anthesis and silking, anthesis-silking interval, plant height, ear height, root lodging, stalk lodging, husk cover, and ear rot, an environment was removed due to low heritability as explained in the methodology.

  1. quality of figures need to improve.

 Response

The quality of the figures has been improved as requested.

Reviewer 2 Report

The manuscript entitled “Genetic Gains in Grain Yield and efficiency of Testing sites of early-maturing maize hybrids under contrasting environments” aimed to study genetic gains of Maize under three different stress environments (Striga infestation, managed drought, and low N). In this study, the authors analyzed genetic gains in grain yields of 44 hybrids during three periods, from 2008 to 2010, from 2011 to 2013, and from 2014 to 2016. Compared to non-stress periods (2017-2019), they found the average rate of yield increase was 84.7 kg/ha/year under stress periods and 65.0 kg/ha/year under non-stress periods. 

Although this study has a lot of interesting data, there are some major concerns about the analysis and conclusion shown below:  

Line 358: It is not clear for the “Genetic gains” of 44 hybrids in the manuscript. It is very important for readers/reviewers to understand grain yields of maize hybrids under different environments. It seems that these hybrids were generated under the selection from 1988 to 2007. Please clarify these details. 

Line 377 - 379: the authors mentioned that the genetic gain and average rate of increase in grain yield was higher compared to 1.33% per year from 1988 to 2007, but didn’t mention the conditions between 1988 and 2007. Were they under stress? Please describe the details. 

Line 412-413: The authors stated that hybrids from period 3 outperformed those from period 1 and period 2. But the figure 1 showed comparative performance of 18 hybrids (No.1 - 18, x-axis), which confused the readers. Please clarify how many hybrids in total are described in figure 1 in the legends and text. Were they 18 hybrids in total? Or totally different hybrids (18 X 3 = 54 hybrids) from period 1, 2, and 3, respectively? 

Line 457-465: In this study, the authors used GGE biplot ( genotype + genotype × environment) to assess the performance and stability of early-maturing maize hybrids. But it is not clear to show the hybrids significantly outperformed (performance and stability) in figure 2. An additional statistical analysis should be carried out if there is a significant interaction between the environment and the genotype to determine the stability level among the 35 genotypes across environments.

Overall, the method used in the study is thorough. But conclusions are not fully supported by the data. Additional statistical analysis is necessary before recommending accepting. 

Author Response

RESPONSES TO REVIEWER’S COMMENTS

 Reviewer 2

Comments and Suggestions for Authors

The manuscript entitled “Genetic Gains in Grain Yield and efficiency of Testing sites of early-maturing maize hybrids under contrasting environments” aimed to study genetic gains of Maize under three different stress environments (Striga infestation, managed drought, and low N). In this study, the authors analyzed genetic gains in grain yields of 44 hybrids during three periods, from 2008 to 2010, from 2011 to 2013, and from 2014 to 2016. Compared to non-stress periods (2017-2019), they found the average rate of yield increase was 84.7 kg/ha/year under stress periods and 65.0 kg/ha/year under non-stress periods. 

Although this study has a lot of interesting data, there are some major concerns about the analysis and conclusion shown below:  

  1. Line 358: It is not clear for the “Genetic gains” of 44 hybrids in the manuscript. It is very important for readers/reviewers to understand grain yields of maize hybrids under different environments. It seems that these hybrids were generated under the selection from 1988 to 2007. Please clarify these details. 

Response

The genetic gains of the individual stress condition have now been provided in the text. Furthermore, the details of the hybrids are described in the materials and methods as follows “A panel of multiple stress-tolerant early yellow and white endosperm hybrids was assembled from the early hybrids developed for high tolerance to Striga, drought, and low N during 9 consecutive years from 2008 to 2016. In total, 54 hybrids developed during three breeding periods (2008–2010; 2011–2013, and 2014–2016) were used for the present study. The hybrids were selected based on their outstanding performance in the regional variety trials in WCA, with several of them sharing the same female parents regardless of the year of origin. Each period consisted of 18 hybrids. Information on the hybrids selected for the present study is shown in Supplementary Table S1”.

  1. Line 377 - 379: the authors mentioned that the genetic gain and average rate of increase in grain yield was higher compared to 1.33% per year from 1988 to 2007, but didn’t mention the conditions between 1988 and 2007. Were they under stress? Please describe the details. 

Response

The details of the conditions between 1988 and 2007 have been included as requested.  “The genetic gain and average rate of increase in grain yield obtained in the present study was higher compared to the 1.33% yr−1 reported by [33]   across Striga parasitism, drought, low soil nitrogen, and optimum conditions. The authors [33] conducted a study at 35 locations in WCA for 2 years to determine genetic improvement in grain yield of open-pollinated varieties developed during  three breeding periods: 1988–2000 (period 1), 2001–2006 (period 2), and 2007–2010 (period 3) whose results were compared to that of the present study.”

  1. Line 412-413: The authors stated that hybrids from period 3 outperformed those from period 1 and period 2. But the figure 1 showed comparative performance of 18 hybrids (No.1 - 18, x-axis), which confused the readers. Please clarify how many hybrids in total are described in figure 1 in the legends and text. Were they 18 hybrids in total? Or totally different hybrids (18 X 3 = 54 hybrids) from period 1, 2, and 3, respectively? 

Response

Each of the numbers 1 - 18 on the x-axis has 3 hybrids (one from each period) comparison represented by the columns. Therefore, a total of 54 hybrids have been compared in Figure 1. As suggested, more details have been provided in the figure legends and in the text.

  1. Line 457-465: In this study, the authors used GGE biplot (genotype + genotype × environment) to assess the performance and stability of early-maturing maize hybrids. But it is not clear to show the hybrids significantly outperformed (performance and stability) in figure 2. An additional statistical analysis should be carried out if there is a significant interaction between the environment and the genotype to determine the stability level among the 35 genotypes across environments.

Response

We have used GGE biplot to decompose the significant interaction effect of G x E observed for grain yield in the analysis of variance results to identify stable and high-yielding hybrids. As explained in the text, the longer the projection of a hybrid onto the single-arrowed line, the lower the stability of the hybrid while the shorter the projections, the greater the stability of the hybrids.

  1. Overall, the method used in the study is thorough. But conclusions are not fully supported by the data. Additional statistical analysis is necessary before recommending accepting. 

 Response

The conclusion has now been modified to be fully supported by the data.

Reviewer 3 Report

Overall, this manuscript authored by Badu-Apraku et al. is well structured and written. It demonstrates a comprehensive study on the evaluation of genetic gains of yield-determining and field traits, along with testing sites suitable for hybrid maize improvement in Africa. The knowledge obtained from this objective-oriented research will benefit the agricultural communities and the society at large. I have a few minor comments as follows.

Table 4 and 5: What are the correlation coefficients between different traits evaluated here? Knowledge of trait correlation could further guide the breeding efforts, for example, the selection of two or multiple traits simultaneously if they are positively correlated.

Table 5: Similar to other tables, the denoted asterisks of the regression coefficients should be introduced.

Line 528-531, 542-546: These sentences within the results section can be condensed to offer a more concise description.

Some sentences in the results and discussion section can be written more concisely.

Author Response

RESPONSES TO REVIEWER’S COMMENTS

Reviewer 3

Overall, this manuscript authored by Badu-Apraku et al. is well structured and written. It demonstrates a comprehensive study on the evaluation of genetic gains of yield-determining and field traits, along with testing sites suitable for hybrid maize improvement in Africa. The knowledge obtained from this objective-oriented research will benefit the agricultural communities and the society at large. I have a few minor comments as follows.

  1. Table 4 and 5: What are the correlation coefficients between different traits evaluated here? Knowledge of trait correlation could further guide the breeding efforts, for example, the selection of two or multiple traits simultaneously if they are positively correlated.

Response

The correlations among the traits used in the present study have been extensively studied and reported in similar studies by Badu-Apraku et al. (2020 & 2022) and we do not see the need to present similar results again.

  1. Table 5: Similar to other tables, the denoted asterisks of the regression coefficients should be introduced.

Response

In Table 5, denoted asterisks on the coefficient of regression (b) for grain yield and other agronomic traits have been introduced.

  1. Line 528-531, 542-546: These sentences within the results section can be condensed to offer a more concise description. Some sentences in the results and discussion section can be written more concisely

Response

The sentences in lines 528-531 and 542-546 within the results section have been condensed to provide a more concise description as suggested.

Reviewer 4 Report

The study with the title “Genetic Gains in Grain Yield and Efficiency of Testing Sites of Early-maturing Maize Hybrids under Contrasting Environments” tested 54 maize hybrids under multiple stresses to identify best performant genotypes for breeding efforts. The stressors tested are those that represent the most limiting for maize yield in SSA.

Authors should verify it the font used is according to template.

Introduction
Very well-written.

Material and Method
sufficiently detailed.

Results
The figures with PCA plots could be made smaller, now are too large.
Tables 4 and 5 are very difficult for the readers to follow comparatively. Authors should think of some better way to present the data from these 2 tables.
The text should not insist excessively on the methodologies nor repeat word by word what is found in tables and figures, only the general trends and more important observations shall be given in text. Make it briefer because the main message is getting lost.

Conclusion
At the end of the introduction authors gave 3 objectives: i), ii), iii), therefore the conclusions should have 3 paragraphs responding to each of those objectives preferably in the same order.

References
Out of 54 sources there are also some from recent years, and some older but all are on the topic.

English quality is OK.

Best regards

small syntax mistakes, minor English style and grammar revision needed

Author Response

RESPONSES TO REVIEWER’S COMMENTS

Reviewer 4

 Results
1. The figures with PCA plots could be made smaller, now are too large.

Response

The figures with PCA plots have been adjusted as suggested

  1. Tables 4 and 5 are very difficult for the readers to follow comparatively. Authors should think of some better way to present the data from these 2 tables.

Response

Tables 4 and 5 have been presented in a more concise, readable, and easily understandable manner to the readers as suggested.

  1. The text should not insist excessively on the methodologies nor repeat word by word what is found in tables and figures, only the general trends and more important observations shall be given in the text. Make it briefer because the main message is getting lost.

Response

The text has been made briefer in areas where necessary.

4.Conclusion
At the end of the introduction authors gave 3 objectives: i), ii), iii), therefore the conclusions should have 3 paragraphs responding to each of those objectives preferably in the same order.

Response

The conclusion has now been changed and has 3 paragraphs responding to each of those objectives in the same order.

Reviewer 5 Report

Dear Authors,

Thank you for submitting this manuscript for publication.

Overall, please explain me the differences and scientific novelty compared to your previous articles published in other journals. Citations number 29, 30, 33, 34.

Analysis and results are good, but I see only a few novelty compared to these previous articles.

Kind regards

English is fine, only minor mistakes are present.

Author Response

RESPONSES TO REVIEWER’S COMMENTS

Reviewer 5

Overall, please explain me the differences and scientific novelty compared to your previous articles published in other journals. Citations number 29, 30, 33, 34.

Response

 The research is novel in the sense that it is the first report on genetic gains in early maize hybrids evaluated under multiple stress conditions. Prior to this report, earlier genetic gains studies had focused on the extraction of open-pollinated varieties. following repeated cycles of recurrent selection. Contrarily, in the present study hybrids developed from inbred lines have been used in estimating the genetic gains obtained after 9 years of hybrid development under multiple stress (Striga infestation, managed drought, and Low N) and non-stress environments.

Round 2

Reviewer 1 Report

the references are not based on MDPI. please revise them.

Reviewer 3 Report

The authors have addressed the concerns that I am aware of.

It is readable, but I recommend some editing to further enhance the cohesion of certain sentences throughout.

Reviewer 4 Report

Dear authors,

I see that issues I brought to your attention were resolved.

Best regards

some fine grammar mistakes

Reviewer 5 Report

Dear Authors,

Thank you for the explanation about your manuscript regarding the novelty, I can accept it.

Kind regards

English is fine, minor grammatical mistakes are still present.